# Sensitive and powerful single-cell RNA sequencing using mcSCRB-seq

Johannes W. Bagnoli [1], Christoph Ziegenhain [1,2], Aleksandar Janjic [1], Lucas E. Wange[1], Beate Vieth[1], Swati Parekh[1,3], Johanna Geuder[1], Ines Hellmann [1] & Wolfgang Enard [1]

Single-cell RNA sequencing (scRNA-seq) has emerged as a central genome-wide method to characterize cellular identities and processes. Consequently, improving its sensitivity, flexibility, and cost-efficiency can advance many research questions. Among the flexible plate-based methods, single-cell RNA barcoding and sequencing (SCRB-seq) is highly sensitive and efficient. Here, we systematically evaluate experimental conditions of this protocol and find that adding polyethylene glycol considerably increases sensitivity by enhancing cDNA synthesis. Furthermore, using Terra polymerase increases efficiency due to a more even cDNA amplification that requires less sequencing of libraries. We combined these and other improvements to develop a scRNA-seq library protocol we call molecular crowding SCRB-seq (mcSCRB-seq), which we show to be one of the most sensitive, efficient, and flexible scRNA-seq methods to date.

[1] Anthropology & Human Genomics, Department of Biology II, Ludwig-Maximilians-University, Großhaderner Straße 2, 82152 Martinsried, Germany. [2] Present address: Department of Cell & Molecular Biology, Karolinska Institutet, 171 77 Stockholm, Sweden. [3] Present address: Max Planck Institute for Biology of Ageing, 50931 Cologne, Germany. These authors contributed equally: Johannes W. Bagnoli, Christoph Ziegenhain, Aleksandar Janjic. Correspondence and requests for materials should be addressed to W.E. (email: enard@bio.lmu.de)

Whole transcriptome single-cell RNA sequencing (scRNA-seq) is a transformative tool with wide applicability to biological and biomedical questions[1,2]. Recently, many scRNA-seq protocols have been developed to overcome the challenge of isolating, reverse transcribing, and amplifying the small amounts of mRNA in single cells to generate high-throughput sequencing libraries[3,4]. However, as there is no optimal, one-size-fits all protocol, various inherent strengths and trade-offs exist[5-7]. Among flexible, plate-based methods, single-cell RNA barcoding and sequencing (SCRB-seq)[8] is one of the most powerful and cost-efficient[6], as it combines good sensitivity, the use of unique molecular identifiers (UMIs) to remove amplification bias and early cell barcodes to reduce costs. Here, we systematically optimize the sensitivity and efficiency of SCRB-seq and generate molecular crowding SCRB-seq (mcSCRB-seq), one of the most powerful and cost-efficient plate-based methods to date (Fig. 1a).

## Results

**Systematic optimization of SCRB-seq.** We started to test improvements to SCRB-seq by optimizing the cDNA yield and quality generated from universal human reference RNA (UHRR)[9] in a standardized SCRB-seq assay (see Supplementary Fig. 1a and Methods). By including the barcoded oligo-dT primers in the lysis buffer, we increased cDNA yield by 10% and avoid a time-consuming pipetting step during the critical phase of the protocol (Supplementary Fig. 1b). Next, we compared the performance of nine Moloney murine leukemia virus (MMLV) reverse transcriptase (RT) enzymes that have the necessary template-switching properties. Especially at input amounts below 100 pg,

Maxima H- (Thermo Fisher) performed best closely followed by SmartScribe (Clontech) (Supplementary Fig. 1c). In order to reduce the costs of the reaction, we showed that cDNA yield and quality is not measurably affected when we reduced the enzyme (Maxima H-) by 20%, reduced the oligo-dT primer by 80%, or used the cheaper unblocked template-switching oligo (Supplementary Fig. 2). Next, we evaluated the effect of $MgCl_2$, betaine and trehalose, as these led to the increased sensitivity of the Smart-seq2 protocol[10]. Since both Smart-seq2 and SCRB-seq generate cDNA by oligo-dT priming, template switching, and PCR amplification, we were surprised that these additives decreased cDNA yield for SCRB-seq (Supplementary Fig. 3a). Apparently, the interactions between enzymes and buffer conditions are complex and optimizations cannot be easily transferred from one protocol to another.

**Molecular crowding significantly increases sensitivity.** An additive that has not yet been explored for scRNA-seq protocols is polyethylene glycol (PEG 8000). It makes ligation reactions more efficient[11] and is thought to increase enzymatic reaction rates by mimicking (macro)molecular crowding, i.e., by reducing the effective reaction volume[12]. As small reaction volumes can increase the sensitivity of scRNA-seq protocols[5,13], we tested whether PEG 8000 can also increase the cDNA yield of SCRB-seq. Indeed, we observed that PEG 8000 increased cDNA yield in a concentration-dependent manner up to tenfold (Supplementary Fig. 3b). However, at higher PEG concentrations, unspecific DNA fragments accumulated in reactions without RNA (Supplementary Fig. 3d) and therefore we chose 7.5% PEG 8000 as an optimal concentration balancing yield and specificity (Supplementary

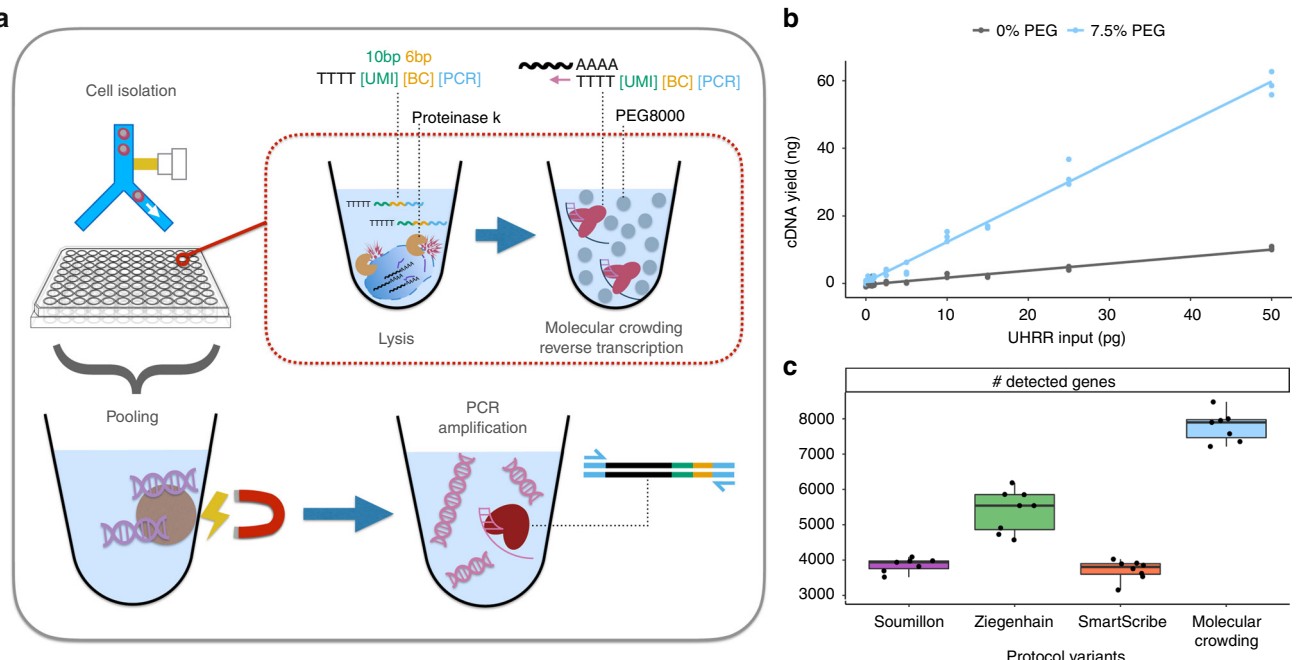

**Fig. 1** mcSCRB-seq workflow and the effect of molecular crowding. **a** Overview of the mcSCRB-seq protocol workflow. Single cells are isolated via FACS in multiwell plates containing lysis buffer, barcoded oligo-dT primers, and Proteinase K. Reverse transcription and template switching are carried out in the presence of 7.5% PEG 8000 to induce molecular crowding conditions. After pooling the barcoded cDNA with magnetic SPRI beads, PCR amplification using Terra polymerase is performed. **b** cDNA yield dependent on the absence (gray) or presence (blue) of 7.5% PEG 8000 during reverse transcription and template switching. Shown are three independent reactions for each input concentration of total standardized RNA (UHRR) and the resulting linear model fit. **c** Number of genes detected (>=1 exonic read) per replicate in RNA-seq libraries, generated from 10 pg of UHRR using four protocol variants (see Supplementary Table 1) at a sequencing depth of one million raw reads. Each dot represents a replicate (n = 8) and each box represents the median and first and third quartiles per method with the whiskers indicating the most extreme data point, which is no more than 1.5 times the length of the box away from the box

Fig. 3c). With the addition of PEG 8000, yield increased substantially, making it possible to detect RNA inputs under 1 pg (Fig. 1b).

To test whether these increases in cDNA yield indeed correspond to increases in sensitivity, we generated and sequenced 32 RNA-seq libraries from 10 pg of total RNA (UHRR) using eight replicates for each of the following four SCRB-seq protocol variants (Supplementary Tables 1, 2): the original SCRB-seq protocol[8] ("Soumillon"; with Maxima H- as RT and Advantage2 as PCR enzyme), the slightly adapted protocol benchmarked in Ziegenhain et al.[6] ("Ziegenhain"; with Maxima H- and KAPA), the same protocol with SmartScribe as the RT enzyme ("SmartScribe") and our optimized protocol ("molecular crowding"; with Maxima H-, KAPA, 7.5% PEG, 80% less oligo-dT, and 20% less Maxima H-). As expected, the molecular crowding protocol yielded the most cDNA, while variant "Soumillon" yielded the least, confirming our systematic optimization (Supplementary Fig. 4a). After sequencing, we processed data using zUMIs[14] and downsampled each of the 32 libraries to one million reads per sample, which has been suggested to correspond to reasonable saturation for single-cell RNA-seq experiments[5,6]. Of the 32 libraries, 31 passed quality control with a median of 71% of the reads mapping to exons (range: 50–77%), 12% to introns (9–15%), 13% to intergenic regions (10–31%), and 4% (3–7%) to no region in the human genome (Supplementary Fig. 4b). Of note, we observe that a higher proportion of reads are mapping to intergenic regions for the "molecular crowding" condition (Supplementary Fig. 4b). As UHRR is provided as DNAse-digested RNA, these reads are likely derived from endogenous transcripts, but why their proportion is increased in the molecular crowding protocol is unclear. In any case, we assessed the sensitivity of the protocols by the number of detected genes per cell (>=1 exonic read), representing a conservative estimate for the molecular crowding protocol with its higher fraction of intergenic reads (Fig. 1c). This sensitivity measure correlates fairly well with cDNA yield (Supplementary Fig. 4a). Hence, it shows that Maxima H- is indeed more sensitive than SmartScribe (5542 detected genes per sample in "Ziegenhain" vs. 3805 in "SmartScribe", $p = 3 \times 10^{-5}$, Welch two sample t-test) and that the molecular crowding protocol is the most sensitive one (7898 vs. 5542 detected genes, $p = 7 \times 10^{-7}$, Welch two sample t-test). In summary, we can show that our optimized SCRB-seq protocol, in particular due to the addition of PEG 8000, increases the sensitivity compared to previous protocol variants at reduced costs.

**Terra retains more complexity during cDNA amplification.** Next, we aimed to increase the efficiency of this protocol by optimizing the cDNA amplification step. Depending on the number of cycles, reaction conditions, and polymerases, substantial noise and bias is introduced when the small amounts of cDNA molecules are amplified by PCR[15,16]. While UMIs allow for the correction of these effects computationally, scRNA-seq methods that have less amplification bias require fewer reads to obtain the same number of UMIs and hence are more efficient[6,17]. As a first step, we evaluated 12 polymerases for cDNA yield and found KAPA, SeqAmp, and Terra to perform best (Supplementary Fig. 5a). We disregarded SeqAmp because of a decreased median length of the amplified cDNA molecules (Supplementary Fig. 5b) as well as the higher cost of the enzyme and continued to compare the amplification bias of KAPA and Terra polymerases. To this end, we sorted 64 single mouse embryonic stem cells (mESCs) and generated cDNA using our optimized molecular crowding protocol. Two pools of cDNA from 32 cells were amplified with KAPA or Terra polymerase (18

cycles) and used to generate libraries. After sequencing and downsampling each transcriptome to one million raw reads[14], we found that amplification using Terra yielded twice as much library complexity (UMIs) than when using KAPA (Supplementary Fig. 5c). This is in agreement with a recent study that optimized the scRNA-seq protocol Quartz-seq2, which also found Terra to retain a higher library complexity[17]. In addition to choosing Terra for cDNA amplification, we also reduced the number of cycles from 19 in the original SCRB-seq protocol to 14, as fewer cycles are expected to decrease amplification bias further[15] and 14 cycles still generated sufficient amounts of cDNA (~1.6–2.4 ng/µl) from mouse ESCs to prepare libraries with Nextera XT (~0.8 ng needed). Depending on the investigated cells, which may have a lower or higher RNA content than ESCs, the cycle number might need to be adapted to generate enough cDNA while avoiding overcycling.

With the final improved version of the molecular crowding protocol (mcSCRB-seq), we tested to what extent cross-contamination occurs. For example, chimeric PCR products may occur following the pooling of cDNA[18] and we assessed whether this might potentially be influenced by PEG that is present during cDNA synthesis before pooling. To this end, we sorted 96 cells of a mixture of mESCs and human-induced pluripotent stem cells, synthesized cDNA according to the mcSCRB-seq protocol with and without the addition of PEG and generated libraries for each of the two conditions. After mapping the sequenced reads to the joint human and mouse reference genomes, each barcode/well could be clearly classified into human or mouse cells, indicating that no doublets were sorted into wells, as may be expected for a fluorescence-activated cell sorting (FACS)-based cell isolation (Supplementary Fig. 6a). Importantly, the median number of reads mapping best to the wrong species is less than 2000 per cell (<0.4% of all reads or <1.5% of uniquely mapped reads). This is not influenced by the addition of PEG, as may be expected, since PEG is only present during cDNA generation (Supplementary Fig. 6b; two-sided t-test, p value = 0.81). In summary, we developed an optimized protocol, mcSCRB-seq, that has higher sensitivity, a less biased amplification and little crosstalk of reads across cells.

**mcSCRB-seq increases sensitivity 2.5-fold more than SCRB-seq.** To directly compare the entire mcSCRB-seq protocol to the previously benchmarked SCRB-seq protocol used in Ziegenhain et al.[6] (Supplementary Table 2), we sorted for each method 48 and 96 single mESCs from one culture into plates, and added ERCC spike-ins[19]. Following sequencing, we filtered cells to discard doublets/dividing cells, broken cells, and failed libraries (see Methods). The remaining 249 high-quality libraries all show a similar mapping distribution with ~50% of reads falling into exonic regions (Supplementary Fig. 7). When plotting the number of detected endogenous mRNAs (UMIs) against sequencing depth, mcSCRB-seq clearly outperforms SCRB-seq and detects 2.5 times as many UMIs per cell at depths above 200,000 reads (Fig. 2a and Supplementary Fig. 8a). At two million reads, mcSCRB-seq detected a median of 102,282 UMIs per cell and a median of 34,760 ERCC molecules, representing 48.9% of all spiked in ERCC molecules (Supplementary Fig. 8b). Assuming that the efficiency of detecting ERCC molecules is representative of the efficiency to detect endogenous mRNAs, the median content per mESC is 227,467 molecules (Supplementary Fig. 8c and 8d), which is very similar to previous estimates using mESCs and STRT-seq, a 5′ tagged UMI-based scRNA-seq protocol[20]. As expected, the higher number of UMIs in mcSCRB-seq also results in a higher number of detected genes. For instance, at 500,000 reads, mcSCRB-seq detected 50,969 UMIs that corresponded to

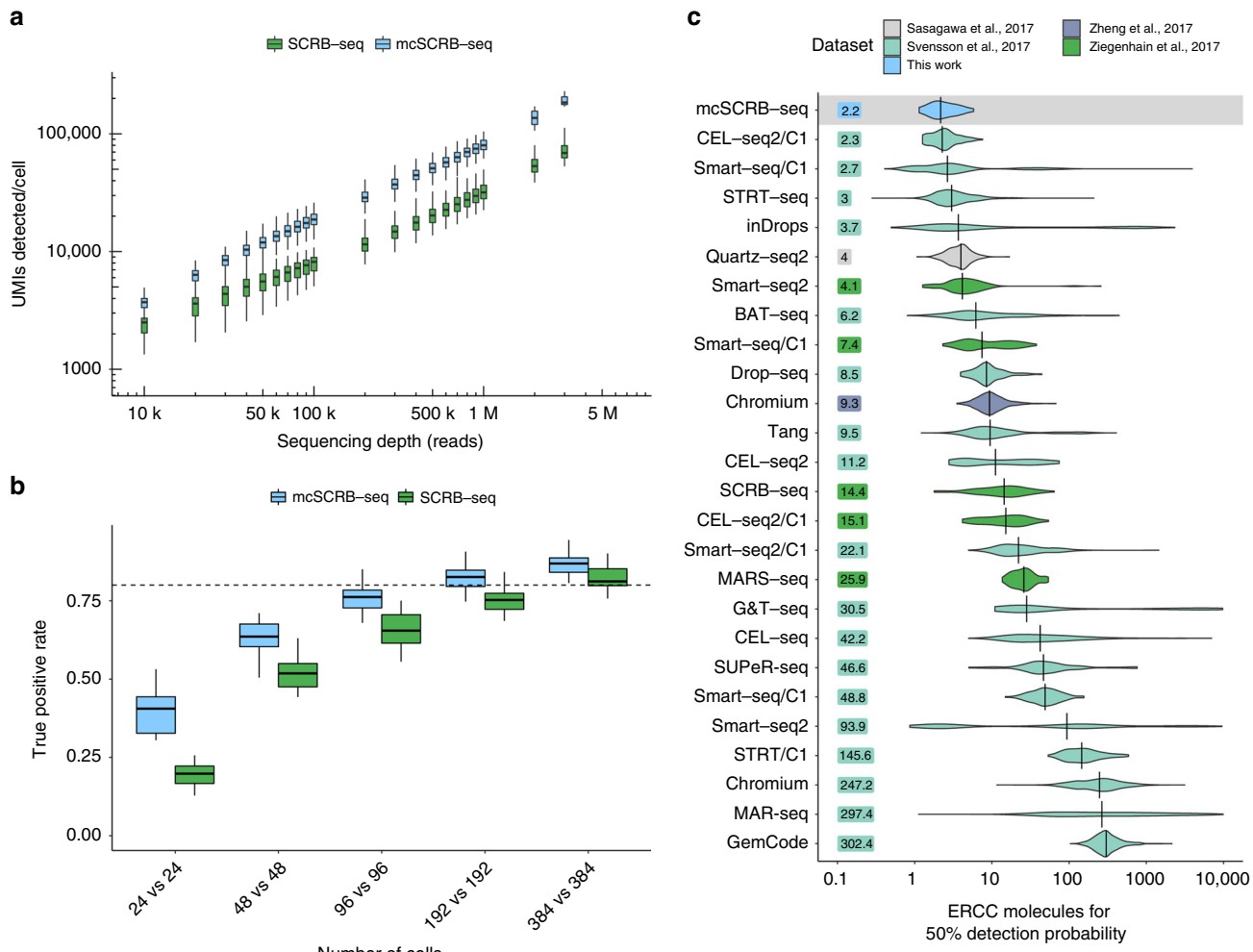

**Fig. 2** Comparison of mcSCRB-seq to SCRB-seq and other protocols. **a** Number of UMIs detected in libraries generated from 249 single mESCs using SCRB-seq or mcSCRB-seq when downsampled to different numbers of raw sequence reads. Each box represents the median and first and third quartiles per cell, sequencing depth and method. Whiskers indicate the most extreme data point that is no more than 1.5 times the length of the box away from the box. **b** The true positive rate of mcSCRB-seq and SCRB-seq estimated by power simulations using the powsimR package[22]. The empirical mean–variance distribution of the 10,904 genes that were detected in at least 10 cells in either mcSCRB-seq or SCRB-seq (500,000 reads) was used to simulate read counts when 10% of the genes are differentially expressed. Boxplots represent the median and first and third quartiles of 25 simulations with whiskers indicating the most extreme data point hat is no more than 1.5 times the length of the box away from the box. The dashed line indicates a true positive rate of 0.8. The matching plot for the false discovery rate is shown in Supplementary Fig. 11d. **c** Sensitivity of mcSCRB-seq and other protocols, calculated as the number of ERCC molecules needed to reach a 50% detection probability as calculated in Svensson et al[5]. Per-cell distributions are shown using violin plots with vertical lines and numbers indicating the median per protocol

5866 different genes, 1000 more than SCRB-seq (Supplementary Fig. 9). Congruent with the above comparison of Terra and KAPA polymerase, mcSCRB-seq showed a less noisy and less-biased amplification (Supplementary Fig. 10). Furthermore, expression levels differed much less between the two batches of mcSCRB-seq libraries, indicating that it could be more robust than SCRB-seq (Supplementary Fig. 11a). In contrast to findings for other protocols[21], neither mcSCRB-seq nor SCRB-seq showed GC content or transcript length-dependent expression levels (Supplementary Fig. 11b, c).

Decisively, we find by using power simulations[6,22] that mcSCRB-seq requires approximately half as many cells as SCRB-seq to detect differentially expressed genes between two groups of cells (Fig. 2b and Supplementary Fig. 11d). Hence, the higher sensitivity and lower noise of mcSCRB-seq compared to SCRB-seq, as measured in parallelly processed cells, indeed matters for quantifying gene expression levels and can be quantified as a doubling of cost-efficiency. Furthermore, we have

reduced the reagent costs from about 1.70 € per cell for SCRB-seq[6] to less than 0.54 € for mcSCRB-seq (Supplementary Fig. 12a and Supplementary Table 3). Together, this makes mcSCRB-seq sixfold more cost-efficient than SCRB-seq. Moreover, owing to an optimized workflow, we could reduce the library preparation time to one working day with minimal hands-on time (Supplementary Fig. 12b and Supplementary Table 4). As SCRB-seq was already one of the most cost-efficient protocols in our recent benchmarking study[6], this likely makes mcSCRB-seq the most cost-efficient plate-based method available.

**Benchmarking by ERCCs.** The widespread use of ERCC spike-ins also allows us to estimate and compare the absolute sensitivity across many scRNA-seq protocols using published data[5]. As in Svensson et al.[5], we used a binomial logistic regression to estimate the number of ERCC transcripts that are needed on average to reach a 50% detection probability (Supplementary Fig. 13a).

mcSCRB-seq reached this threshold with 2.2 molecules, when ERCCs are sequenced to saturation (Supplementary Fig. 13b). When comparing this to a total of 26 estimates for 20 different protocols obtained from two major protocol comparisons[5,6] as well as additional relevant protocols[17,23], mcSCRB-seq has the highest sensitivity among all protocols compared to date (Fig. 2c). It should be noted that the data show large amounts of variation within protocols, even for well-established, sensitive methods like Smart-seq2. This is the case, especially in Svensson et al.[5], because the data were generated from many varying cell types sequenced in numerous labs. Similarly, mcSCRB-seq sensitivity estimates could be variable across labs and conditions. Nevertheless, the average ERCC detection efficiency is the most representative measure to compare sensitivities across many protocols.

**mcSCRB-seq detects biological differences in complex tissues.** Finally, we applied mcSCRB-seq to peripheral blood mononuclear cells (PBMCs), a complex cell population with low mRNA amounts, to test whether it is efficient in recapitulating biological differences. We obtained PBMCs from one healthy donor, FACS-sorted cells in four 96-well plates and prepared libraries using mcSCRB-seq with a more stringent lysis condition (see Methods; Fig. 3a). We sequenced ~203 million reads for the resulting pool, of which ~189 million passed filtering criteria in the *zUMIs* pipeline (see Methods). Next, we filtered low-quality cells (<50,000 raw reads or mapping rates <75%; Supplementary Fig. 14a), leaving 349 high-quality cells for further analysis (Supplementary Fig. 14b). Using the Seurat package[24], we clustered the expression data and obtained five clusters that could be easily attributed to expected cell types: B cells, Monocytes, NK cells, and T cells (Fig. 3b). Rare cell types, such as dendritic cells or megakaryocytes that are known to occur in PBMCs at frequencies of ~0.5–1%, could not be detected, as expected from the low power to cluster 2–3 cells. For the detected cell types, known marker gene expression fits closely to previously described results[23] (Fig. 3c, d). Overall, we show that mcSCRB-seq is a powerful tool to highlight biological differences, already when a low number of cells are sequenced.

## Discussion

In this work, we developed mcSCRB-seq, a scRNA-seq protocol utilizing molecular crowding. Based on benchmarking data generated from mouse ES cells, we show that mcSCRB-seq considerably increases sensitivity and decreases amplification bias due to the addition of PEG 8000 and the use of Terra polymerase, respectively. Furthermore, it shows no indication of bias for GC content and transcript lengths, and has low levels of crosstalk between cell barcodes, which has been seen especially in droplet-based RNA-seq approaches[23,25]. Compared to the previous SCRB-seq protocol, mcSCRB-seq increases the power to quantify gene expression twofold. Additionally, optimized reagents and workflows reduce costs by a factor of three. Qualitatively, we validate our protocol by sequencing PBMCs, a complex mixture of different cell types. We show that mcSCRB-seq can identify the different subpopulations and marker gene expression correctly and distinctively detect the major cell types present in the population.

In this context, we found that it was necessary to use different lysis conditions for the PBMCs than for mESCs. In our experience, some cell types may require a more stringent lysis buffer to stabilize mRNA, which might be a result of internal RNAses and/or lower RNA content. Therefore, we also provide an alternative lysis strategy for mcSCRB-seq to deal with more difficult cell types or samples.

Taken together, mcSCRB-seq is—to the best of our knowledge—not only the most sensitive protocol when benchmarked using ERCCs, it is also the most cost-efficient and flexible plate-based protocol currently available, and could be a valuable methodological addition to many laboratories, in particular as it requires no specialized equipment and reagents.

## Methods

**cDNA yield assay.** For all optimization experiments, universal human reference RNA (UHRR; Agilent) was utilized to exclude biological variability. Unless otherwise noted, 1 ng of UHRR was used as input per replicate. Additionally, Proteinase K digestion and desiccation were not necessary prior to reverse transcription. In order to accommodate all the reagents, the total volume for reverse transcription was increased to 10 μl. All concentrations were kept the same, with the exception that we added the same total amount of reverse transcriptase (25 U), thus lowering the concentration from 12.5 to 2.5 U/μl. After reverse transcription, no pooling was performed, rather preamplification was done per replicate. For each sample, we measured the cDNA concentration using the Quant-iT PicoGreen dsDNA Assay Kit (Thermo Fisher).

**Comparison of reverse transcriptases.** Nine reverse transcriptases, Maxima H- (Thermo Fisher), SMARTScribe (Clontech), Revert Aid (Thermo Fisher), Enz-Script (Biozym), ProtoScript II (New England Biolabs), Superscript II (Thermo Fisher), GoScript (Promega), Revert UP II (Biozym), and M-MLV Point Mutant (Promega), were compared to determine which enzyme yielded the most cDNA. Several dilutions ranging from 1 to 1000 pg of universal human reference RNA (UHRR; Agilent) were used as input for the RT reactions.

RT reactions contained final concentrations of 1 × M-MuLV reaction buffer (NEB), 1 mM dNTPs (Thermo Fisher), 1 μM E3V6NEXT barcoded oligo-dT primer (IDT), and 1 μM E5V6NEXT template-switching oligo (IDT). For reverse transcriptases with unknown buffer conditions, the provided proprietary buffers were used. Reverse transcriptases were added for a final amount of 25 U per reaction.

All reactions were amplified using 25 PCR cycles to be able to detect low inputs.

**Comparison of template-switching oligos (TSO).** Unblocked (IDT) and blocked (Eurogentec) template-switching oligonucleotides were compared to determine yield when reverse transcribing 10 pg UHRR and primer-dimer formation without UHRR input. Reaction conditions for RT and PCR were as described above.

**Effect of reaction enhancers.** In order to improve the efficiency of the RT, we tested the addition of reaction enhancers, including $MgCl_2$, betaine, trehalose, and polyethylene glycol (PEG 8000). The final reaction volume of 10 μl was maintained by adjusting the volume of $H_2O$.

For this, we added increasing concentrations of $MgCl_2$ (3, 6, 9, and 12 mM; Sigma-Aldrich) in the RT buffer in the presence or absence of 1 M betaine (Sigma-Aldrich). Furthermore, the addition of 1 M betaine and 0.6 M trehalose (Sigma-Aldrich) was compared to the standard RT protocol. Lastly, increasing concentrations of PEG 8000 (0, 3, 6, 9, 12, and 15% W/V) were also tested.

**Comparison of PCR DNA polymerases.** The following 12 DNA polymerases were evaluated in preamplification: KAPA HiFi HotStart (KAPA Biosystems), SeqAmp (Clontech), Terra direct (Clontech), Platinum SuperFi (Thermo Fisher), Precisor (Biocat), Advantage2 (Clontech), AccuPrime Taq (Invitrogen), Phusion Flash (Thermo Fisher), AccuStart (QuantaBio), PicoMaxx (Agilent), FideliTaq (Affymetrix), and Q5 (New England Biolabs). For each enzyme, at least three replicates of 1 ng UHRR were reverse transcribed using the optimized molecular crowding reverse transcription in 10 μl reactions. Optimal concentrations for dNTPs, reaction buffer, stabilizers, and enzyme were determined using the manufacturer's recommendations. For all amplification reactions, we used the original SCRB-seq PCR cycling conditions[8].

**Cell culture of mouse embryonic stem cells.** J1[26] and JM8[27] mouse embryonic stem cells (mESCs) were provided by the Leonhardt lab (LMU Munich) and originally provided by Kerry Tucker (Ruprecht-Karls-University,Heidelberg) and by the European Mouse Mutant Cell repository (JM8A3; www.eummcr.org), respectively. They were used for the comparison of KAPA vs. Terra PCR amplification (Supplementary Fig. 5c) and the comparison of SCRB-seq and mcSCRB-seq, respectively. Both were cultured under feeder-free conditions on gelatine-coated dishes in high-glucose Dulbecco's modified Eagle's medium (Thermo Fisher) supplemented with 15% fetal bovine serum (FBS, Thermo Fisher), 100 U/ml penicillin, 100 μg/ml streptomycin (Thermo Fisher), 2 mM L-glutamine (Thermo Fisher), 1 × MEM non-essential amino acids (NEAA, Thermo Fisher), 0.1 mM β-mercaptoethanol (Thermo Fisher), 1000 U/ml recombinant mouse LIF (Merck Millipore) and 2i (1 μM PD032591 and 3 μM CHIR99021 (Sigma-Aldrich)). mESCs were routinely passaged using 0.25% trypsin (Thermo Fisher).

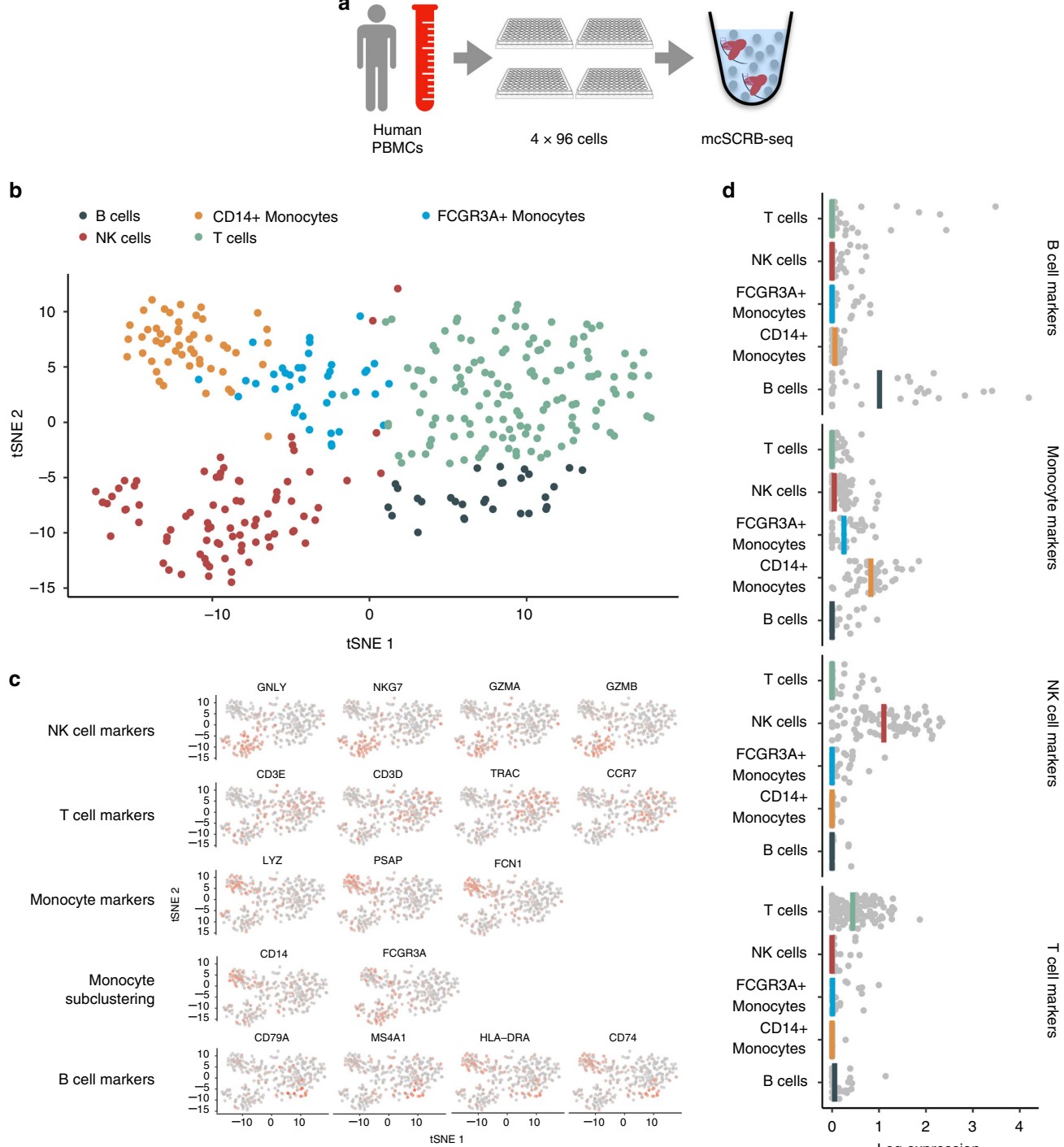

**Fig. 3** mcSCRB-seq distinguishes cell types of peripheral blood mononuclear cells. **a** PBMCs were obtained from a healthy male donor and FACS sorted into four 96-well plates. Using the mcSCRB-seq protocol, sequencing libraries were generated. **b** tSNE projection of PBMC cells (n = 349) that were grouped into five clusters using the Seurat package[24]. Colors denote cluster identity. **c** tSNE projection of PBMC cells (n = 349) where each cell is colored according to its expression level of various marker genes for the indicated cell types. Expression levels were log-normalized using the Seurat package. **d** Marker gene expression from **c** was summarized as the mean log-normalized expression level per cell. B-cell markers: *CD79A, CD74, MS4A1, HLA-DRA*; Monocyte markers: *LYZ, PSAP, FCN1, CD14, FCGR3A*; NK-cell markers: *GNLY, NKG7, GZMA, GZMB*; T-cell markers: *CD3E, CD3D, TRAC, CCR7*

mESC cultures were confirmed to be free of mycoplasma contamination by a PCR-based test[28].

**Cell culture of human-induced pluripotent stem cells**. Human-induced pluripotent stem cells were generated using standard techniques from renal epithelial cells obtained from a healthy donor with written informed consent in accordance with the ethical standards of the responsible committee on human experimentation (216–08, Ethikkommission LMU München) and with the

current (2013) version of the Declaration of Helsinki. hiPSCs were cultured under feeder-free conditions on Geltrex (Thermo Fisher)-coated dishes in StemFit medium (Ajinomoto) supplemented with 100 ng/ml recombinant human basic FGF (Peprotech) and 100 U/ml penicillin, 100 μg/ml streptomycin (Thermo Fisher). Cells were routinely passaged using 0.5 mM EDTA. Whenever cells were dissociated into single cells using 0.5 × TrypLE Select (Thermo Fisher), the culture medium was supplemented with 10 μM Rho-associated kinase (ROCK) inhibitor Y27632 (BIOZOL) to prevent apoptosis.

hiPSC cultures were confirmed to be free of mycoplasma contamination by a PCR-based test[28].

**SCRB-seq cDNA synthesis.** Cells were dissociated using trypsin and resuspended in 100 µl of RNAprotect Cell Reagent (Qiagen) per 100,000 cells. Directly prior to FACS sorting, the cell suspension was diluted with PBS (Gibco). Single cells were sorted into 96-well DNA LoBind plates (Eppendorf) containing lysis buffer using a Sony SH800 sorter (Sony Biotechnology; 100 µm chip) in "Single Cell (3 Drops)" purity. Lysis buffer consisted of a 1:500 dilution of Phusion HF buffer (New England Biolabs). After sorting, plates were spun down and frozen at −80 °C. Libraries were prepared as previously described[6,8]. Briefly, proteins were digested with Proteinase K (Ambion) followed by desiccation to inactivate Proteinase K and reduce the reaction volume. RNA was then reverse transcribed in a 2 µl reaction at 42 °C for 90 min. Unincorporated barcode primers were digested using Exonuclease I (Thermo Fisher). cDNA was pooled using the Clean & Concentrator-5 kit (Zymo Research) and PCR amplified with the KAPA HiFi HotStart polymerase (KAPA Biosystems) in 50 µl reaction volumes.

**mcSCRB-seq cDNA synthesis.** A full step-by-step protocol for mcSCRB-seq has been deposited in the protocols.io repository[29]. Briefly, cells were dissociated using trypsin and resuspended in PBS. Single cells ("3 drops" purity mode) were sorted into 96-well DNA LoBind plates (Eppendorf) containing 5 µl lysis buffer using a Sony SH800 sorter (Sony Biotechnology; 100 µm chip). Lysis buffer consisted of a 1:500 dilution of Phusion HF buffer (New England Biolabs), 1.25 µg/µl Proteinase K (Clontech), and 0.4 µM barcoded oligo-dT primer (E3V6NEXT, IDT). After sorting, plates were immediately spun down and frozen at −80 °C. For libraries containing ERCCs, 0.1 µl of 1:80,000 dilution of ERCC spike-in Mix 1 was used.

Before library preparation, proteins were digested by incubation at 50 °C for 10 min. Proteinase K was then heat inactivated for 10 min at 80 °C. Next, 5 µl reverse transcription master mix consisting of 20 units Maxima H- enzyme (Thermo Fisher), 2× Maxima H- Buffer (Thermo Fisher), 2 mM each dNTPs (Thermo Fisher), 4 µM template-switching oligo (IDT), and 15% PEG 8000 (Sigma-Aldrich) was dispensed per well. cDNA synthesis and template switching was performed for 90 min at 42 °C. Barcoded cDNA was then pooled in 2 ml DNA LoBind tubes (Eppendorf) and cleaned up using SPRI beads. Purified cDNA was eluted in 17 µl and residual primers digested with Exonuclease I (Thermo Fisher) for 20 min at 37 °C. After heat inactivation for 10 min at 80 °C, 30 µl PCR master mix consisting of 1.25 U Terra direct polymerase (Clontech) 1.66 × Terra direct buffer and 0.33 µM SINGV6 primer (IDT) was added. PCR was cycled as given: 3 min at 98 °C for initial denaturation followed by 15 cycles of 15 s at 98 °C, 30 s at 65 °C, 4 min at 68 °C. Final elongation was performed for 10 min at 72 °C.

**Library preparation.** Following preamplification, all samples were purified using SPRI beads at a ratio of 1:0.8 with a final elution in 10 µl of H₂O (Invitrogen). The cDNA was then quantified using the Quant-iT PicoGreen dsDNA Assay Kit (Thermo Fisher). Size distributions were checked on high-sensitivity DNA chips (Agilent Bioanalyzer). Samples passing the quantity and quality controls were used to construct Nextera XT libraries from 0.8 ng of preamplified cDNA.

During library PCR, 3′ ends were enriched with a custom P5 primer (P5NEXTPT5, IDT). Libraries were pooled and size-selected using 2% E-Gel Agarose EX Gels (Life Technologies), cut out in the range of 300–800 bp, and extracted using the MinElute Kit (Qiagen) according to manufacturer's recommendations.

**Sequencing.** Libraries were paired-end sequenced on high output flow cells of an Illumina HiSeq 1500 instrument. Sixteen bases were sequenced with the first read to obtain cellular and molecular barcodes and 50 bases were sequenced in the second read into the cDNA fragment. When several libraries were multiplexed on sequencing lanes, an additional 8 base i7 barcode read was done.

**Primary data processing.** All raw fastq data were processed using zUMIs together with STAR to efficiently generate expression profiles for barcoded UMI data[14,30]. For UHRR experiments, we mapped to the human reference genome (hg38) while mouse cells were mapped to the mouse genome (mm10) concatenated with the ERCC reference. Gene annotations were obtained from Ensembl (GRCh38.84 or GRCm38.75). Downsampling to fixed numbers of raw sequencing reads per cell were performed using the "-d" option in zUMIs.

**Filtering of scRNA-seq libraries.** After initial data processing, we filtered cells by excluding doublets and identifying failed libraries. For doublet identification, we plotted distributions of total numbers of detected UMIs per cell, where doublets were readily identifiable as multiples of the major peak.

In order to discard broken cells and failed libraries, spearman rank correlations of expression values were constructed in an all-to-all matrix. We then plotted the distribution of "nearest-neighbor" correlations, i.e., the highest observed correlation value per cell. Here, low-quality libraries had visibly lower correlations than average cells.

**Species-mixing experiment.** Mouse ES cells (JM8) and human iPS cells were mixed and sorted into a 96-well plate containing lysis buffer as described for mcSCRB-seq using a Sony SH800 sorter (Sony Biotechnology; 100 µm chip). cDNA was synthesized according to the mcSCRB-seq protocol (see above), but without addition of PEG 8000 for half of the plate. Wells containing or lacking PEG were pooled and amplified separately. Sequencing and primary data analysis was performed as described above with the following changes: cDNA reads were mapped against a combined reference genome (hg38 and mm10) and only reads with unique alignments were considered for expression profiling.

**Complex tissue analysis.** PBMCs were obtained from a healthy male donor with written informed consent in accordance with the ethical standards of the responsible committee on human experimentation (216–08, Ethikkommission LMU München) and with the current (2013) version of the Declaration of Helsinki. Cells were sorted into 96-well plates containing 5 µl lysis buffer using a Sony SH800 sorter (Sony Biotechnology; 100 µm chip). Lysis buffer consisted of 5 M Guanidine hydrochloride (Sigma-Aldrich), 1% 2-mercaptoethanol (Sigma-Aldrich) and a 1:500 dilution of Phusion HF buffer (New England Biolabs). Before library preparation, each well was cleaned up using SPRI beads and resuspended in a mix of 5 µl reverse transcription master mix (see above) and 4 µl ddH₂O. After the addition of 1 µl 2 µM barcoded oligo-dT primer (E3V6NEXT, IDT), cDNA was synthesized according to the mcSCRB-seq protocol (see above). Pooling was performed by adding SPRI bead buffer. Sequencing and primary data analysis was performed as described above using the human reference genome (hg38). We retained only high-quality cells with at least 50,000 reads and a mapping rate above 75%. Furthermore, we discarded potential doublets that contained more than 40,000 UMIs and 5000 genes. Next, we used Seurat[24] to perform normalization (LogNormalize) and scaling. We selected the most variable genes using the "FindVariableGenes" command (1108 genes). Next, we performed dimensionality reduction with PCA and selected components with significant variance using the "JackStraw" algorithm. Statistically significant components were used for shared nearest-neighbor clustering (FindClusters) and tSNE visualization (RunTSNE). Log-normalized expression values were used to plot marker genes.

**Estimation of cellular mRNA content.** For the estimation of cellular mRNA content in mESCs, we utilized the known total amount of ERCC spike-in molecules added per cell. First, we calculated a detection efficiency as the fraction of detected ERCC molecules by dividing UMI counts to total spiked ERCC molecule counts. Next, dividing the total number of detected cellular UMI counts by the detection efficiency yields the number of estimated total mRNA molecules per cell.

**ERCC analysis.** In order to estimate sensitivity from ERCC spike-in data, we modeled the probability of detection in relation to the number of spiked molecules. An ERCC transcript was considered detected from 1 UMI. For each cell, we fitted a binomial logistic regression model to the detection of ERCC genes given their input molecule numbers. Using the MASS R-package, we determined the molecule number necessary for 50% detection probability.

For public data from Svensson et al.[5], we used their published molecular abundances calculated using the same logistic regression model obtained from Supplementary Table 2 (https://www.nature.com/nmeth/journal/v14/n4/extref/nmeth.4220-S3.csv). For Quartz-seq2[17], we obtained expression values for ERCCs from Gene Expression Omnibus (GEO; GSE99866), sample GSM2656466; for Chromium[23] we obtained expression tables from the 10× Genomics webpage (https://support.10xgenomics.com/single-cell-gene-expression/datasets/1.1.0/ercc) and for SCRB-seq, Smart-seq2, CEL-seq2/C1, MARS-seq and Smart-seq/C1[6], we obtained count tables from GEO (GSE75790). For these methods, we calculated molecular detection limits given their published ERCC dilution factors.

**Power simulations.** For power simulation studies, we used the powsimR package[22]. Parameter estimation of the negative binomial distribution was done using scran normalized counts at 500,000 raw reads per cell[31]. Next, we simulated two-group comparisons with 10% differentially expressed genes. Log2 fold-changes were drawn from a normal distribution with a mean of 0 and a standard deviation of 1.5. In each of the 25 simulation iterations, we draw equal sample sizes of 24, 48, 96, 192 and 384 cells per group and test for differential expression using ROTS[32] and scran normalization[31].

**Batch effect analysis.** In order to detect genes differing between batches of one scRNA-seq protocol, data were normalized using scran[31]. Next, we tested for differentially expressed genes using limma-voom[33,34]. Genes were labeled as significantly differentially expressed between batches with Benjamini–Hochberg adjusted p values <0.01.

**Code availability.** Analysis code to reproduce major analyses can be found at https://github.com/cziegenhain/Bagnoli_2017.

**Data availability.** RNA-seq data generated here are available at GEO under accession GSE103568.

Further data including cDNA yield of optimization experiments is available on GitHub (https://github.com/cziegenhain/Bagnoli_2017). A detailed step-by-step protocol for mcSCRB-seq has been submitted to the protocols.io repository (mcSCRB-seq protocol 2018). All other data available from the authors upon reasonable request.

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

## Acknowledgements

We thank Ines Bliesener for expert technical assistance. We are grateful to Magali Soumillon and Tarjei Mikkelsen for providing the original SCRB-seq protocol and to Stefan Krebs and Helmut Blum for sequencing. We would like to thank Elena Winheim for the PBMC sample. This work was supported by the Deutsche Forschungsgemeinschaft (DFG) through LMUexcellent and the SFB1243 (Subproject A14/A15).

## Author contributions

C.Z. and W.E. conceived the study. J.W.B., C.Z., A.J. and L.E.W. performed experiments and prepared sequencing libraries. J.G. and J.W.B. cultured mouse ES and human iPS cells. Sequencing data were processed by S.P. and C.Z. J.W.B., C.Z., A.J. and B.V. analyzed the data. J.W.B., C.Z., A.J., I.H. and W.E. wrote the manuscript.

## Additional information

**Competing interests:** The authors declare no competing interests.

