## [Peer Review File · Nature Communications]

Reviewers' comments:

Reviewer #1 (Remarks to the Author):

The manuscript by Bagnoli et al. describes a modified approach of the SCRIB-seq method to perform single-cell RNA-seq. This approach based on the principle defined as molecular crowding greatly enhances the quality of single-cell libraries. The authors did an admirable job by systematically testing different combinations and analyzing the sequencing statistics. This approach has the potentialities to become a major technology in the field.

One crucial test to do is to test whether the increased number of detected genes increases the power of biological analysis. It seems that the mcSCRIB-seq was performed on mESCs so it would be a great opportunity to compare this dataset with Ziegenhain et al. 2017 beyond what has been done in figure 2C. One possible analysis would be to test whether mcSCRIB-seq can better detect crucial mESC genes involved in pluripotency, priming and differentiation and/or whether enhanced single cell information can help distinguishing more subpopulations in the mESC culture.

Minor comments:

- The side-by-side comparisons in figure 2C show the performance of mcSCRIB-seq, but at the same time show how variable single-cell methods are. Even a very robust approach like Smart-seq/C1 shows up to 1 log difference between labs. It would be worth discussing the reasons for this variability (i.e., cell source or technical variability) even in the light that the mcSCRIB approach might not perform as well as in the developers' lab.
- Can the author describe in S4B the lower capture of exons and the higher capture of intronic and intergenic regions? Are these non-exonic regions transcribed?

Reviewer #2 (Remarks to the Author):

In this manuscript, Bagnoli et al. described the use of a most sensitive, cost-efficient and flexible plate-based protocol currently available for scRNA-Seq (mcSCRIB-seq). They improved and evaluated previously published scRNA-seq method "Single-Cell RNA-Barcoding and Sequencing" (SCRIB-seq). The sensitivity and efficiency of the protocol was systematically optimized by comparing different reverse transcription enzymes, DNA polymerases, titrating reagents/oligos and exploring a new additive (PEG 8000) in RT that increased a cDNA yield substantially.

This is a nice method manuscript, and we only have a few minor points:

(1) The authors reported the development of the mcSCRIB-seq technique that is the most sensitive scRNA-seq methods to date. However, the recent article (Ziegenhain et al. Mol. Cell 65, 631–643.e4) published by the same authors stated that the most sensitive method is the Smart-seq2 protocol. Taking into consideration the fact that there is no parallel comparison of these two protocols it would be correct to say that mcSCRIB-seq is most sensitive among 3' UMI counting based scRNA-seq methods.

(2) The authors first tested the cDNA yield using different MMLVs and showed Maxima H- and SmartScribe were the most efficient. However, in later section, only Maxima H- was used in the molecular crowding protocol. SmartScribe was not tested under the mcSCRIB protocol. Is there any particular reason for this? Is it due to price/cost difference? It will be useful to provide a direct comparison of Maxima H- and SmartScribe under the same mcSCRIB protocol.

(3) The authors showed PEG concentration increased desired cDNA yield, but unspecific cDNA yield was also increased. To show the current mcSCRIB protocol amplifies desired cDNA, the authors should state the percentage of reads mapped to the transcriptomes in the main text. It took a while to locate this information in Supplementary Fig 4b.

(4) In Supplementary Figure 1C, Maxima H- was coloured as blue, and the rest grey. Are they tested in different conditions (based on Supp Fig 1B, blue means oligo-dT in lysis and grey means oligo-dT in RT)? This was a little bit misleading.

(5) Due to the nature of the paper it would be helpful for a reader to have more data about RT enzymes titration for Maxima and Smartscribe and also for PCR cycles titration for Terra and HiFi, i.e. how the PCR cycle number was determined? Was qPCR performed to make sure the current cycle number does not reach saturation?

Reviewer #3 (Remarks to the Author):

The authors present an optimized version of SCRB-Seq, deemed mSCRB-seq, which improves sensitivity and lowers costs for plate based scRNA-seq experiments. The major result is the impressive improvement of yield by adding PEG to the RT reaction. They also test PCR enzymes and identify an alternative to KAPA with lower cost, and potentially more even amplification.

We were intrigued when we saw this manuscript on biorXiv, and while we don't use SCRB-seq, my lab did test the idea of adding PEG to the RT reaction. We did see a consistent 2-3 fold improvement (not the 10-fold the authors saw, but still impressive). Given that the addition of PEG costs essentially nothing, this is a very neat result with the potential to boost sensitivities across scRNA-seq protocols, not just SCRB-seq. We did not see any improvements (either in sequencing or in yield) from the use of Terra, but this may be system dependent.

A few comments:

1. The authors should claim a 3, not 6-fold, reduction in cost. They add the extra 2-fold since they claim that one needs to sequence 1/2 as many cells, but this is obviously very tailored to one specific experiment.
2. It is essential that the authors produce a human mouse plot, showing that the increase in yield is not due to the possibility of increased contamination between cells after pooling. This can manifest as increased sensitivity. In the species mixing plot, the authors must calculate not only the doublet rate, but also the species contamination rate of the singlets (i.e. the slope of the lines along the axes), and compare to the same protocol without addition of PEG.
3. The authors should test this protocol not just on cell lines, but on complex cell populations. I would suggest PBMCs, which contain small T cells that are challenging to get sensitive data for. Showing clear clustering between these populations again would boost confidence that there is no technical artifacts driving the increased sensitivity.

Reviewers' comments:

Reviewer #1 (Remarks to the Author):

The manuscript by Bagnoli et al. describes a modified approach of the SCR-seq method to perform single-cell RNA-seq. This approach based on the principle defined as molecular crowding greatly enhances the quality of single-cell libraries. The authors did an admirable job by systematically testing different combinations and analyzing the sequencing statistics. This approach has the potentialities to become a major technology in the field.

One crucial test to do is to test whether the increased number of detected genes increases the power of biological analysis. It seems that the mcSCR-seq was performed on mESCs so it would be a great opportunity to compare this dataset with Ziegenhain et al. 2017 beyond what has been done in figure 2C. One possible analysis would be to test whether mcSCR-seq can better detect crucial mESC genes involved in pluripotency, priming and differentiation and/or whether enhanced single cell information can help distinguishing more subpopulations in the mESC culture.

We would like to thank the reviewer for the positive review of our manuscript. Biological analysis based on mESCs was not informative because their culture condition (2i/LIF) was chosen to minimize biological variation for our benchmarking. As shown in Kolodziejczyk et al., 2015, 2i/LIF-cultured mESCs contain only one transcriptional subpopulation (2C-like cells) that occur at a frequency of just ~3%. With the number of mESCs in our dataset, we would expect about four 2C-like cells for SCR-seq/mcSCR-seq. Thus, we opted to generate new data to showcase the power of mcSCR-seq to answer biological questions. As noted in the response to reviewer 3 (see below comment for more details), we sequenced PBMCs and show clustering of the major cell types contained in this cell population. The results of this additional data are presented in a new Figure 3.

Minor comments:

- The side-by-side comparisons in figure 2C show the performance of mcSCR-seq, but at the same time show how variable single-cell methods are. Even a very robust approach like Smart-seq/C1 shows up to 1 log difference between labs. It would be worth discussing the reasons for this variability (i.e., cell source or technical variability) even in the light that the mcSCR approach might not perform as well as in the developers' lab.

This is a valid and important observation. We have included a discussion of the variability of sensitivity estimates in the relevant section of the main text.

- Can the author describe in S4B the lower capture of exons and the higher capture of intronic and intergenic regions? Are these non-exonic regions transcribed?

Indeed, the higher fraction of intergenic regions observed in the "molecular crowding" condition must represent transcribed non-exonic regions, because libraries were constructed from a commercially available DNase-digested purified RNA (universal human reference RNA). We added this reasoning to the main text of the paper and also mention that this is specific to UHRR, as a higher fraction of intergenic reads is not observed in the SCR-seq vs mcSCR-seq mouse ES cell comparison (Supplementary Fig. 7).

Reviewer #2 (Remarks to the Author):

In this manuscript, Bagnoli et al. described the use of a most sensitive, cost-efficient and flexible plate-based protocol currently available for scRNA-Seq (mcSCRB-seq). They improved and evaluated previously published scRNA-seq method “Single-Cell RNA-Barcoding and Sequencing” (SCRB-seq). The sensitivity and efficiency of the protocol was systematically optimized by comparing different reverse transcription enzymes, DNA polymerases, titrating reagents/oligos and exploring a new additive (PEG 8000) in RT that increased a cDNA yield substantially.

This is a nice method manuscript, and we only have a few minor points:

(1) The authors reported the development of the mcSCRB-seq technique that is the most sensitive scRNA-seq methods to date. However, the recent article (Ziegenhain et al. Mol. Cell 65, 631–643.e4) published by the same authors stated that the most sensitive method is the Smart-seq2 protocol. Taking into consideration the fact that there is no parallel comparison of these two protocols it would be correct to say that mcSCRB-seq is most sensitive among 3' UMI counting based scRNA-seq methods.

Based on the comparison using ERCCs as shown in Figure 2C, mcSCRB-seq is more sensitive (2.2 molecules for 50% detection probability) than Smart-seq2 as performed in Ziegenhain et al (4.1 molecules) that is in turn more sensitive than SCRB-seq as performed in Ziegenhain et al (14.4 molecules). Thus, based on this ERCC comparison we think that the improvements presented here lead to mcSCRB-seq being the most sensitive method overall, not only among 3' UMI methods. We thus state “Hence, mcSCRB-seq is [...] the most sensitive protocol when benchmarked using ERCCs”.

(2) The authors first tested the cDNA yield using different MMLVs and showed Maxima H- and SmartScribe were the most efficient. However, in later section, only Maxima H- was used in the molecular crowding protocol. SmartScribe was not tested under the mcSCRB protocol. Is there any particular reason for this? Is it due to price/cost difference? It will be useful to provide a direct comparison of Maxima H- and SmartScribe under the same mcSCRB protocol.

This was actually not correctly described by us. The result of our enzyme comparison was that Maxima H- is the best enzyme followed by SmartScribe based on cDNA yield (Figure S1C). We then directly compared these two by sequencing UHRR libraries and again find maxima H- to clearly outperform SmartScribe (Figure 1C: Ziegenhain et al 2017 vs. SmartScribe). Hence, we performed the molecular crowding protocol with Maxima. In addition, the list price of SmartScribe is about 70% higher making the enzyme even less attractive. We have now made this rationale clearer in the main text.

(3) The authors showed PEG concentration increased desired cDNA yield, but unspecific cDNA yield was also increased. To show the current mcSCRB protocol amplifies desired cDNA, the authors should state the percentage of reads mapped to the transcriptomes in the main text. It took a while to locate this information in Supplementary Fig 4b.

We now state the percentage of unmapped reads and reads mapping to exons, introns, intergenic regions in the main text. In any case, we assess the sensitivity for desired cDNA by the number of detected genes per one million raw reads, which confirms that the higher cDNA yields indeed does lead to a higher relevant sensitivity.

(4) In Supplementary Figure 1C, Maxima H- was coloured as blue, and the rest grey. Are they tested in different conditions (based on Supp Fig 1B, blue means oligo-dT in lysis and grey means oligo-dT in RT)? This was a little bit misleading.

The color code we use is made to highlight those conditions selected for the final mcSCRB-seq protocol in blue as opposed to all the other evaluated conditions (grey/teal). Because this was not obvious previously, we added the information on the color code to the relevant figures.

(5) Due to the nature of the paper it would be helpful for a reader to have more data about RT enzymes titration for Maxima and Smartscribe and also for PCR cycles titration for Terra and HiFi, i.e. how the PCR cycle number was determined? Was qPCR performed to make sure the current cycle number does not reach saturation?

All major data from this paper, also the reverse transcriptase enzyme titration, is freely available in our GitHub repository. We have added this comment to the "data availability" statement.

We have not systematically optimized the number of PCR cycles because we were working with our familiar mESC system. We describe now in the main text that we aim for cDNA concentrations that are approx. 2-3 times above as needed for fragmentation by Nextera XT (ie > 3 x 0.8 ng) and mention that this needs to be adapted when working with cells that have less or more RNA content than mESCs.

Reviewer #3 (Remarks to the Author):

The authors present an optimized version of SCRБ-Seq, deemed mSCRБ-seq, which improves sensitivity and lowers costs for plate based scRNA-seq experiments. The major result is the impressive improvement of yield by adding PEG to the RT reaction. They also test PCR enzymes and identify an alternative to KAPA with lower cost, and potentially more even amplification.

We were intrigued when we saw this manuscript on biorXiv, and while we don't use SCRБ-seq, my lab did test the idea of adding PEG to the RT reaction. We did see a consistent 2-3 fold improvement (not the 10-fold the authors saw, but still impressive). Given that the addition of PEG costs essentially nothing, this is a very neat result with the potential to boost sensitivities across scRNA-seq protocols, not just SCRБ-seq. We did not see any improvements (either in sequencing or in yield) from the use of Terra, but this may be system dependent.

A few comments:

1. The authors should claim a 3, not 6-fold, reduction in cost. They add the extra 2-fold since they claim that one needs to sequence 1/2 as many cells, but this is obviously very tailored to one specific experiment.

We have now tried to clarify this calculation and its rationale in the text and give it a more informal try here:

The increased sensitivity and reduced amplification bias of mcSCRБ-seq compared to SCRБ-seq increases the power (=detecting truly differentially expressed genes at the same FDR) by 2-fold, i.e. for the same information half the amount of cells (hence money) is needed. This does not depend on the specific experiment as it is based on the estimated mean and variance across genes as measured when single cell are processed with the mcSCRБ-seq and the SCRБ-seq protocols. As the same cells were processed in parallel with these two protocols, differences in mean and variance across genes are caused by differences in the two protocols. The simulations are needed to quantify how this mean-variance distribution across genes actually affects the quantification of expression levels

(assessed by simulating and detecting differentially expressed genes). Without these power simulations it is just not clear/quantifiable how much improvements to a protocol actually help. As we find that mcSCRB-seq can quantify gene expression levels as good as SCRIB-seq with just half the number of cells, we quantify this improvement as a 2-fold cost-reduction, which then can also be multiplied with the 3-fold reduction in reagents costs per cell. While we do not want to stress the exact cost reduction, we think that this “bang for the buck” approach and rationale is really crucial to compare the utility of methods as we have done already in our previous benchmarking paper.

2. It is essential that the authors produce a human mouse plot, showing that the increase in yield is not due to the possibility of increased contamination between cells after pooling. This can manifest as increased sensitivity. In the species mixing plot, the authors must calculate not only the doublet rate, but also the species contamination rate of the singlets (i.e. the slope of the lines along the axes), and compare to the same protocol without addition of PEG.

This is a very valid point and to address it we sequenced a mixture of human iPS and mouse ES cells using the mcSCRB-seq protocol in the presence and absence of PEG. The results are presented in Supplementary Figure 6 and show that contamination is not an issue neither with nor without PEG: While doublets were not observed at all (Figure S6A), as maybe expected for a FACS-based cell isolation, less than 2000 reads per cell (<0.4% of all reads or <1.5% of uniquely mapped reads) are derived from the “wrong” species independent of PEG ($p=0.81$; Figure S6A). Hence, the increased sensitivity due to PEG is clearly not caused by increased cross-contamination after pooling.

3. The authors should test this protocol not just on cell lines, but on complex cell populations. I would suggest PBMCs, which contain small T cells that are challenging to get sensitive data for. Showing clear clustering between these populations again would boost confidence that there is no technical artifacts driving the increased sensitivity.

Following the suggestion of the reviewer, we used PBMCs as a complex example population. We prepared a library from four 96-well plates using the mcSCRB-seq protocol and sequenced it to an average depth of ~400,000 reads per cell. This experiment confirms the good quality data produced by mcSCRB-seq, with over 90% of cells passing quality control and sensitive data with an average of ~2500 genes detected. We can show clear clustering of the main cell types of PBMCs (T-cells, B-cells, NK-cells and Monocytes), demonstrating the utility of mcSCRB-seq also for complex cell populations. We display these new results in the new Figure 3 and Supplementary Figure 14.

REVIEWERS' COMMENTS:

Reviewer #1 (Remarks to the Author):

This is a great technical manuscript. No further revision needed.

Reviewer #2 (Remarks to the Author):

This reviewer thanks the authors for the detailed address of all issues posed by all reviewers. The current manuscript assessed several important aspects of scRNA-seq protocols, and will be a valuable resource for the community. It is good for publication in Nature Communication.

Reviewer #3 (Remarks to the Author):

The authors have addressed my concerns, and I was able to reproduce their analysis on the PBMC dataset. I recommend the manuscript be published.